# Viral Metagenomics for Identification of Emerging Viruses in Transfusion Medicine

**DOI:** 10.3390/v14112448

**Published:** 2022-11-04

**Authors:** Svetoslav Nanev Slavov

**Affiliations:** 1Department of Cellular and Molecular Therapy (NuCeL), Butantan Institute, São Paulo 05503-900, SP, Brazil; svetoslav.slavov@butantan.gov.br; Tel.: +55-(16)-2101-9300 (ext. 9365); 2Laboratory of Bioinformatics, Blood Center of Ribeirão Preto, Faculty of Medicine of Ribeirão Preto, University of São Paulo, Rua Tenente Catão Roxo 2501, Ribeirão Preto CEP 14051-140, SP, Brazil

**Keywords:** next-generation sequencing, viral metagenomics, hemotherapy, emerging viruses, blood donation

## Abstract

Viral metagenomics has revolutionized our understanding for identification of unknown or poorly characterized viruses. For that reason, metagenomic studies gave been largely applied for virus discovery in a wide variety of clinical samples, including blood specimens. The emerging blood-transmitted virus infections represent important problem for public health, and the emergence of HIV in the 1980s is an example for the vulnerability of Blood Donation systems to such infections. When viral metagenomics is applied to blood samples, it can give a complete overview of the viral nucleic acid abundance, also named “blood virome”. Detailed characterization of the blood virome of healthy donors could identify unknown (emerging) viral genomes that might be assumed as hypothetic transfusion threats. However, it is impossible only by application of viral metagenomics to assign that one viral agent could impact blood transfusion. That said, this is a complex issue and will depend on the ability of the infectious agent to cause clinically important infection in blood recipients, the viral stability in blood derivatives and the presence of infectious viruses in blood, making possible its transmission by transfusion. This brief review summarizes information regarding the blood donor virome and some important challenges for use of viral metagenomics in hemotherapy for identification of transfusion-transmitted viruses.

## 1. Introduction

Viral metagenomics, which reveals the viral abundance in a wide variety of samples using next generation sequencing (NGS), is currently one of the fastest growing scientific disciplines. The advances of metagenomics have been related to improvements in NGS and bioinformatic approaches used for the analysis of the generated sequence read data [1]. The continuous application and improvements of metagenomics have shown that the viral abundance of clinical samples is much greater than previously suspected and, currently, many viruses and their genotypes are being identified and annotated [2].

An important utilization of viral metagenomics is the identification of unknown viruses in clinical specimens obtained from symptomatic patients where the current methods fail to establish a diagnostic conclusion. NGS procedures have already been used for identification of clinically important emerging viruses, including bunyaviruses [3], Brazilian mammarenaviruses [4], Lassa virus outbreak [5] and emerging gastrointestinal viruses [6]. The success of viral discovery has also been attributed to the potency of sequencers used to generate extraordinary sequencing depth. The Illumina ultimate generation sequencers (also named production scale sequencers) like NextSeq 1000/2000 and NovaSeq 6000 have the capacity to generate a maximum of 1.2 billion sequences (360 gb) for the NextSeq 1000/2000 platform and 20 billion reads (6000 gb) for Illumina NovaSeq 6000, respectively. The high depth of sequencing permits the identification of viruses with low representation in the tested samples, especially the emerging ones (Figure 1).

The generation of a high number of sequence data has also impacted the bioinformatic analysis. Such analysis requires high computational capacity and specialized workflow represented by bioinformatic pipelines able to systematically classify large quantities of reads of variable size (short reads, Illumina; long reads, nanopore technologies). However, it is not always that the taxonomic classification of generated reads is straightforward; for many of them, there might be no reference genomes in the public databases. The bioinformatic workflow for metagenomic analysis might represent a challenge for researchers with limited knowledge in programming languages and computer science [7].

## 2. Viral Metagenomics and Blood Transfusion

The application of NGS and metagenomic analysis on blood samples could provide insights for the viral nucleic acid abundance in such specimens. In this respect, the high potential of metagenomics for virus discovery can represent a suitable means for identification of poorly described or emerging viruses in blood donors or hemoderivatives that could be regarded as possible transfusion threats.

Infectious diseases have played an important role in blood transfusion, and historically, HCV, HBV and especially HIV have greatly impacted the routine diagnostic tests and blood donor selection criteria [8]. Currently, due to the sensitive diagnosis, the residual risk for transfusion-transmission of these infections is invariably very low [9], even in HIV highly endemic countries like those in Sub-Saharan Africa [10,11]. However, the identification of transfusion-transmitted viral infections with a short viremic phase and preferentially transmitted by arthropod vectors showed that exotic viruses can also impact hemotherapy and cause clinical impact in recipients. These infections include several arboviruses, transmitted mosquitos, ticks or sandflies, which, during large outbreaks, cause a high percentage of asymptomatic infections related to contamination of donated blood. The most notorious example includes Dengue viruses 1–4 (DENV 1–4), causing Dengue fever, which is the most important arbovirus disease in the tropics. During large outbreaks, asymptomatically DENV-infected blood donors might be a cause for contamination of donated blood products [12,13]. Clinical complications due to transfusion-transmitted DENV, especially in patients with underlying diseases have also been reported [14,15,16]. Transfusion-transmission of other arboviruses related to clinical consequences were reported for the Japanese encephalitis virus (JEV), where the blood recipient developed severe encephalitis and death [17] and for Ross River virus, where the transfusion-transmission was related to a symptomatic infection [18]. ZIKV transfusion-transmission has also been reported but with no apparent clinical effects [19].

Given this, the question arises: What is the best technique for identification of other possible emerging viruses that could show transfusion-transmission? Due to the unbiased character of sequence detection, viral metagenomics applied to blood donors or hemoderivatives is the most suitable technique for the discovery of unknown or unsuspected viruses in blood products [20]. In the best possible case, a viral metagenomic survey might be applied among blood donors from “hotposts” or geographic areas with accelerated viral emergence or presence of a high number of uncharacterized zoonotic viruses (Figure 1). These are by no means the tropical and equatorial regions of the world, where the superpopulation, pronounced variety of animal species and transmitting vectors, closer human–animal contact and precarious sanitary conditions can favor the emergence of novel viruses. Some of these infections, under specific circumstances, could enter in the blood donation cycle [21,22].

## 3. The Human Blood Virome

The human blood virome can be defined as the community of all viruses found in blood specimens of healthy individuals or patients. The normal component of the human blood virome includes mainly commensal viruses like a high variety of anelloviruses, Human pegivirus-1 (HPgV-1), possibly Human pegivirus-2 (HPgV-2) and probably many other unidentified viruses which are regarded as non-pathogenic [20,23,24]. Using metagenomics viruses that do not make part of this community might be also identified. They may have emerging character and, due to their identification in blood, concerns regarding their transfusion-transmission might be raised. As a convenience, this section will be subdivided into “commensal component of the human virome” and “pathogenic viruses identified in blood donors by metagenomics”.

### 3.1. The Commensal Blood Donor Virome

The most abundant viruses of the human blood virome are the anelloviruses [25]. Metagenomics on Chinese healthy blood donors with elevated ALT/AST levels showed mainly anellovirus types and HPgV-1 and no emerging viruses. The same viruses were observed in healthy blood donors with no alterations of the hepatic enzymes. This was the same in healthy blood donors with no alterations of the hepatic enzymes [26]. Spanish blood donors showed similar results [23,27]; with exclusive detection, human commensal viruses have been detected. The extensive presence of commensal viruses in eligible blood donors can be considered a possible reflection of the efficient donor selection process, including sensitive routine diagnostic tests. These strategies are probably responsible for the low diversity of pathogenic viruses among eligible blood donors identified by metagenomics.

The anelloviruses are classified into three genera: Alphatorqueteno virus (different Torque teno virus species, TTV), betatorqueviruses (species of the torque teno mini viruses, TTMV) and gammatorqueviruses (different species of the Torque teno midi viruses, TTMDV). Their total percentage in human blood reaches 97% and that is why the virome of healthy blood donors could also be also called “anellovirome” [23,27]. Anelloviruses show extreme genetic diversity. Due to frequent NGS application, many new genomic forms and genotypes are continuously being discovered [28,29]. Despite the cosmopolitan distribution of the anelloviruses, they are regarded as non-pathogenic [30]. Anellovirus bloom, both in diversity and quantitative abundance, is documented in conditions related to immune suppression, especially in patients with HIV or organ/stem cell transplantation [31,32,33]. For that reason, anellovirus viral load has been suggested as biomarkers of immune suppression [34]. Despite that anelloviruses are inoffensive for the healthy population, their impact might be different in blood recipients with altered immune functions. In that line, the identification of a novel TTMV type in patients with Hodgkin lymphoma [29], although probably accidental, may suggest that more detailed studies are needed to evaluate if anelloviruses might exert pathogenic effect.

A less frequently described viral component of the healthy blood donor virome is HPgV-1. This is probably justified by the lower prevalence of HPgV-1 in the blood donor population, globally reaching 3.1% [35]. HPgV-1 is also a commensal virus without specific clinical importance. Nevertheless, there is evidence that HPgV-1 in coinfection with HIV could beneficially modify the cellular immune response thus slowing the progression to AIDS [36,37]. Another non-pathogenic human pegivirus, HPgV-2, with similar to genomic organization to HPgV-1, was also described in 2015. Although its prevalence is still unclear, it seems that it is infrequent in the healthy population [24]. Further metagenomic studies will reveal if HPgV-2 is also a normal component of the blood virome.

The healthy blood virome also shows DNA traces of papillomaviruses [38] and Merkel cell polyomavirus [39]. The presence of these dermatological oncoviruses in the virome of eligible blood donors is regarded as a contamination originating from the vein puncture used for blood donation. Other circular viruses like gemycircularviruses and gemykibiviruses that probably also make part of the human healthy virome [38,40] are with currently unknown impact for blood transfusion.

### 3.2. In Search of Pathogenic Transfusion-Transmitted Viruses: Metagenomics Applied on Blood Donors Reporting Postdonation Illness

Postdonation illness (PDI) is characterized by the abrupt appearance of acute infectious symptoms in otherwise healthy blood donors after donation and who call back to the blood donation institution reporting this occurrence [41]. In such cases, the application of viral metagenomics on these samples can give valuable information in respect of the PDI causative agents and is a suitable route to identify viruses that could impact transfusion safety.

Brazilian metagenomic studies were performed on plasma samples from blood donors reporting PDI symptoms including fever, retro-orbital pain, exanthema, headache, myalgia, diarrhea, respiratory symptoms and jaundice. Viruses like DENV-2, parvovirus B19 (B19V) and even influenza virus (H3N2) were identified by viral metagenomics in several samples [41,42] and subsequently confirmed by molecular techniques. In this respect, application of metagenomics in blood donors who consequently presented symptomatic diseases reveals the potential of metagenomics to unveil viruses with potential impact on transfusion safety and shows the importance of PDI for prevention of transfusion-transmission of infectious agents that are not routinely tested by the Blood Transfusion Services.

An important observation is that the above cited studies were performed in tropical regions, where many arboviral and emerging viruses intensively circulate. Ideally, a global multicentric metagenomic study including blood donors who report PDI from different climatic areas, is the best option to unveil a broad variety of viruses capable of causing symptoms in blood donors and with probable transfusion-transmission.

## 4. Challenges of the Metagenomic Analysis in Regards of Blood Donation

When viral metagenomics is applied to blood donor samples in search of emerging viruses, it can meet several challenges that could be grouped in the following categories: (i) sample preparation; (ii) bioinformatic analysis; and (iii) impact of the identified viral genomes on transfusion safety.

Viral metagenomics is related to multiple laboratorial challenges, especially sample preparation. However, one of the most important is the high presence of contaminating host nucleic acids in the extracted samples that can significantly impact the sensitivity of viral identification. For such reason, host-depletion and viral concentration techniques (filtration, ultracentrifugation, addition of nucleic acid carriers) must be mandatory applied during laboratory procedures with samples destined for metagenomics [43]. Although such depletion/concentration procedures have beneficial impact on host nucleic acids removal from clinical samples, they could also lead to the diminishment of viral genomes of interest reflecting the preparation of sequencing libraries [43]. The presence of host nucleic acids represents significant challenge in the metagenomics. Therefore, different laboratory techniques that are directed towards diminishment of host genetic material must be comprehensively compared, and ones providing the best ratio between host nucleic acid decontamination and viral genome acid preservation might be further established as laboratory standards for sample management.

During metagenomic processing, the sensitive preparation techniques may introduce viral sequences originating from the reagents regarded as contaminants and artifacts (Figure 1). Reagents used for extraction of nucleic acids, concentration of viral particles, reverse transcription, amplification and library preparations contain viruses or viral products that could be identified in the subsequent bioinformatic analysis. A classic example is the reverse transcriptase derived from the Moloney murine leukemia virus and used for cDNA synthesis, that might introduce genomic material belonging to this virus that can be further identified by bioinformatic analysis. Our practice shows frequent presence of *Shamonda* (orthobunyavirus sequences) and Simu viral reads that are thought to be derived from the RQ1 DNAse present in kits used for next-generation sequencing [44]. Another frequently observed contaminant in our analysis is the BeAn 58,058 virus, belonging to the Poxviridae family, which is naturally isolated from *Oryzomys* sp. rodents in Brazil [45]. All these sequence reads also probably originate from laboratory kits used in the sample preparation.

Contaminants may also originate from surfaces, clinical and laboratory equipment, laboratory staff and sources that enter in contact with the clinical sample during preparation. These contaminants are regarded as external [46]. Depending on the laboratory where the samples were processed, in our practice, we also observed frequent contaminations with external reads originating from laboratory surfaces, for example metagenomic identification of SARS-CoV-2 reads in archived samples collected long before the emergence of this virus, when the sample processing was performed in laboratories dealing with SARS-CoV-2 genomic surveillance. Cross-contamination due to manipulation, named internal contamination, may also lead to misleading results, especially when in the sequence run there are samples with high viral load. Internal contamination could also result from PCR errors and index switching [46]. In our practice, the most frequently observed internal contamination is due to cross-contamination between the samples, but it can be easily solutioned by targeted viral testing in the individual samples.

The so-called “computational contaminants” could also affect the metagenomic analysis. The presence of small fragments of human DNA contaminants in assembled microbial sequences has been reported. This finding makes challengeable the application of metagenomics for the identification of infectious agents in human samples, once the bioinformatic analysis compares the obtained from human specimen sequence reads to known viral, bacterial, parasitic or fungal sequences that might be implied as cause of infection. The exclusion of contaminant reads and identification of the cause of infection in such situations is fundamental to avoid misdiagnosis [47].

The contaminating sequences have specific features that could provide valuable clues in their identification. They tend to appear stochastically and, during assembly, do not form larger contigs. Moreover, they might be presented as short, identical, regionally repeated reads that probably result from the low loads present in the tested samples. A simple method to identify viral contaminants is to perform a BLAST search that will almost always result in cloning or expression vectors. Viruses with animal origin that are identified in patient samples should be also handled with caution and regarded as environmental contaminants or taxonomic errors as discussed below. To reduce the risk of contamination and to improve the interpretation in metagenomic analysis, several controls, including blank and negative, should be introduced in each sequencing [44].

Each laboratory uses different bioinformatic pipelines for the analysis of the generated data that also can generate discordant results, even when are applied on the same sample. The use of different alignment software and taxonomic classifiers shows variable precision and accuracy, especially when targeting low number of sequence reads. The best strategy to increase the precision and accuracy of taxonomy is the use of combination of several taxonomic classifiers. For example, the combination of different classifiers could increase the precision to over 95% [48]. Another important issue is the availability of updated classification databases with adequate viral abundance. The almost universal presence of high percentages of unclassified reads (regarded also as “dark matter”) in the metagenomic datasets is an indicator of the impossibility of the currently available taxonomic classifiers to deal with extreme diversity of reads, most of which probably belong to unknown microorganisms. In such cases, virus discovery adopts “de novo” assembly of unclassified reads and examines their phylogenetic relationships based on genomic similarity.

Most of the currently used next-generation sequencers produce short reads (typically 50–300 bp) that might also be a cause for imprecise classification. To overcome this problem, sequencers generating longer reads, especially utilizing nanopore mechanism, are desirable [48]. Nevertheless, although nanopore sequencing has been widely applied for viral genomic surveillance [49,50,51], its use for metagenomics is still doubtful, mainly due to the modest coverage that might miss low viral load infections and lack of suitable metagenomic protocols. For this reason, strategies to increase sequence length and introduce well-optimized protocols for viral metagenomics should be regarded as priority by the NGS companies and scientific community.

Even more challenging is establishing transfusion-transmission properties of the identified by metagenomics viral agents in blood samples (Figure 1). Even if we assume that a genome of unknown virus is assembled and taxonomically classified, multiple studies must be performed until this agent could be classified as a potential emerging transfusion threat. For example, viral nucleic acids of respiratory viruses like influenza A [42] and SARS-CoV-2 [52] have been detected in plasma samples from blood donors, but their transfusion-transmission is probably ineffective [53], and as a result their importance for the transfusion medicine is limited.

In order for a virus genome identified by metagenomics to be considered transfusion-transmitted, it must respond to several issues. One of the most important criteria is that the virus must cause a recognizable clinical impact in recipients. For example, the anelloviruses and HPgV-1 that are a normal component of the human virome are frequently transmitted by blood transfusion. However, they cause no apparent clinical disease in blood recipients. Another important characteristic is that the virus must remain viable in collected blood and must be infectious when introduced by vein punction. Also of importance is the frequency with which the infection is transmitted to susceptible recipients (transfusion-transmission rates), which depends on the duration of viremia and blood donation frequency. Potential transfusion-transmitted viruses must also be evaluated in regard of viremia kinetics, immune responses and window periods. [54]. It is obvious that, in its current form, the viral metagenomics cannot respond to any of the abovementioned criteria; however, a rapid metagenomic identification could suggest viral agents that could be further recognized as possible transfusion-transmission threats.

## 5. Final Remarks and Conclusions

The appearance of unsuspected viruses has greatly impacted the transfusion policies and services worldwide. The HIV emergence in the early 1980s is a classic example underlining the vulnerability of the transfusion system to emerging viral threats [55]. HIV was characterized within three years, demanding international cooperation and significant efforts. Nowadays, with the advent of NGS and bioinformatics, the genomic characterization of a given viral agent in clinical materials takes hours. Therefore, in the near future, NGS could be regarded as the method of choice for molecular diagnosis, vaccine development and new therapeutic strategies [56].

Viral metagenomics has also revolutionized the field of hemotherapy and especially blood-borne infections. Currently, at this stage, viral metagenomics is best applied for the identification of unknown or poorly characterized agents. Nevertheless, implying transfusion-transmission potential of such viruses is difficult due to challenges related to sample preparation (mainly viral concentration, abundance of host nucleic acids) and the bioinformatic profile of the blood virome (presence of commensal viruses and contaminants among others). Moreover, due to the cost, laborious preparation and analysis of such reactions, viral metagenomics is restricted to specific situations where a transfusion-transmission of an emerging agent is highly suspected rather than routine use in the hemotherapy practice. However, in tropical areas or regions with high potential of explosive viral outbreaks and/or virus emergence that could impact blood donor population, viral metagenomics can be applied in terms of identification of potential viruses that might impact blood transfusion process, especially considering the experience with DENV or ZIKV during outbreaks.

It is of utmost importance to provide the safest blood products that are free of infectious agents. The revolutionary development of NGS and bioinformatic analysis in the face of metagenomics has also been applied to blood donor samples and has thus been able to reveal their genetic content including all viruses present. This has raised questions regarding the presence of emerging viral agents that could impact transfusion safety. In that line, metagenomics can be successfully used as a rapid and efficient approach to identify a viral genome especially when the virus is poorly described or completely unknown. Although discovery of viral nucleic acids could shape our opinion for future transfusion threats, it is not sufficient to imply that one genetic sequence could have transfusion-transmission properties. Several issues, the most important of which include the clinical impact of the virus in the recipient, viral survival in the stored blood, viral load capable of causing infection and efficiency of virus transmissibility by blood, must be thoroughly investigated. This argues against the routine application of viral metagenomics as a tool for viral infectious surveillance in the Blood Banks worldwide. In that line, risk-based decision making for blood safety should be the most important tool combing health economics and operational assessments before introduction of any NGS procedure [57]. Although viral metagenomics could give us insights on the possible emerging transfusion threats, it is insufficient only by genomic characterization to reveal viruses that can impact transfusion medicine. The balance between economic costs and the most important health priorities of given country will further shape the possible application of viral metagenomics as a routine tool for identification of emerging viruses in the field of blood transfusion.

## Figures and Tables

**Figure 1 viruses-14-02448-f001:**
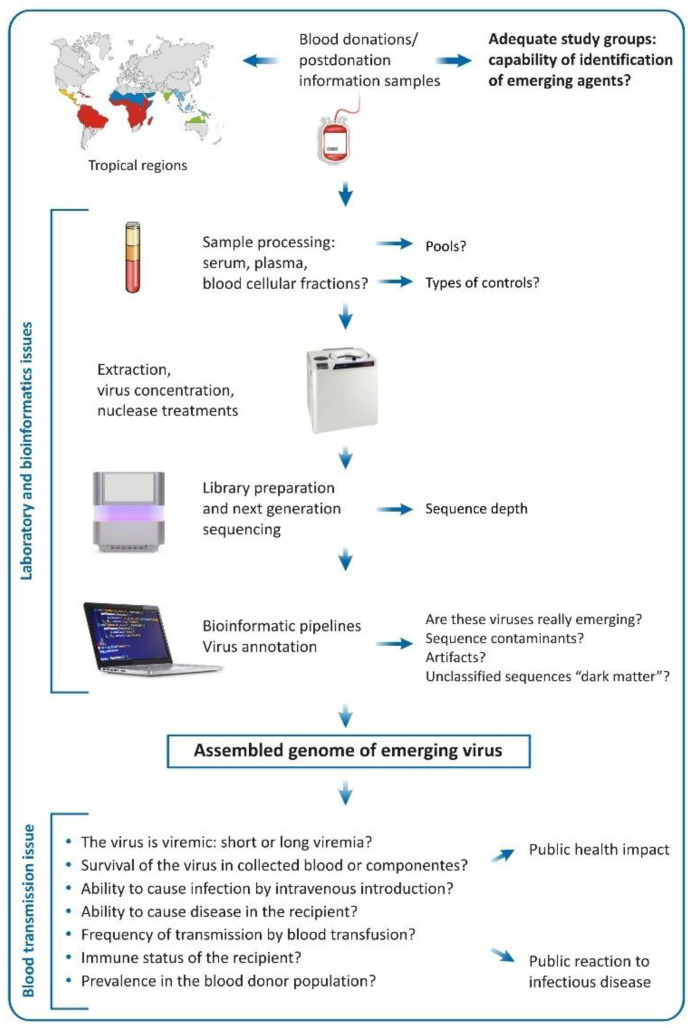
Schematic workflow (pipeline) for identification of emerging viruses in blood donations. The definition of the target study group is important, whereas eligible blood donors will be tested or blood donors reporting some kind of symptoms (postdonation illness). The study location is also of importance; the tropical regions are “hotspots” for the emergence of viruses and therefore are more suitable for metagenomic studies applied to search of emerging agents in blood donation. Many issues regarding sample processing need to be answered, such the assembly of the samples in pools, types of controls, concentration of viral particles, beyond the type of sequencer that will be used. After sequencing, the bioinformatic analysis will reveal the viral composition (abundance) of the samples, but caution must be taken especially regarding contaminants, artifacts and the unclassified sequences. Even if we assume that, in the tested samples, there is an emerging virus, many other questions must be answered in order to be regarded as a possible transfusion threat (survival in blood donations, transmissibility via intravenous route, ability to cause disease in the recipient, levels and lasting of viremia, frequency of transmission by transfusion, among others).

## Data Availability

Not applicable.

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
