# Peer review of "Viral Metagenomics for Identification of Emerging Viruses in Transfusion Medicine"

_viruses, 2022, doi:10.3390/v14112448_

Round 1
Reviewer 1 Report
Title: Viral Metagenomics Applied to Blood Donation. In this review article, Dr. Slavov summarized the concepts, methods, and applications of viral metagenomics with a focus on transfusion safety. The review did not bring us new information, knowledge, and clear future direction. However, I think it remains valuable for the beginners in the field or readers from other fields.
Although there is a raid growth in publication, serum/plasma viral metagenomics has many problems in terms of technical aspects. One of major barriers is the lack of a robust method that can enrich viral sequences. Virus has a tiny genome so that clinical specimens like serum/plasma is dominated by host nucleic acids. Consequently, viral metagenomics has a low sensitivity. This issue was raised in previous review articles, for example, Houldcroft and Breuer, Nat. Rev Microbiology 2017. However, it remains unsolved. Capture sequencing could increase the sensitivity but it works only for known viruses. Increase of sequencing depth could be a partial solution but it makes the contamination worse. There are reports of other methods such as enzyme treatment and ultracentrifuge. However, we don’t know if these methods really work because of the lack of comparison. While this review manuscript is specially for blood donation, the dominance of host nucleic acid in serum/plasma is a root cause of NGS detection sensitivity and contamination. Therefore, I suggest Dr. Slavov to mention this important point in appropriate sections, for example, section 4 (Challenges …).
Minor: Proof reading may be necessary. For example, line 267: “….fom the low viral loads of ……..” should be “…..from the load loads of …….” Because contaminants are not necessarily viruses.
Author Response
RESPONSE TO THE REVIEWER COMMENTS/CRITICISMS
REVIEWER #1
Title: Viral Metagenomics Applied to Blood Donation. In this review article, Dr. Slavov summarized the concepts, methods, and applications of viral metagenomics with a focus on transfusion safety. The review did not bring us new information, knowledge, and clear future direction. However, I think it remains valuable for the beginners in the field or readers from other fields.
I am grateful to the reviewer for the positive feedback regarding the submitted brief review. In fact, the presented manuscript is a summary of the available literature regarding the human blood donor virome (commensal and pathogenic viruses that can be found by metagenomics) and also tries to encompass the most important challenges related to the interpretation of the metagenomic data, especially in the light of transfusion transmitted viruses. The manuscript was thoroughly modified, the conclusions and the final remarks now try to give more insights on the applicability of the viral metagenomics for discovery of emerging viruses that might impact transfusion safety. The suggestions of the reviewer were very valuable and they helped to improve the presented information in the manuscript and the future directions of metagenomcs in the field of blood transfusion.
Although there is a raid growth in publication, serum/plasma viral metagenomics has many problems in terms of technical aspects. One of major barriers is the lack of a robust method that can enrich viral sequences. Virus has a tiny genome so that clinical specimens like serum/plasma is dominated by host nucleic acids. Consequently, viral metagenomics has a low sensitivity. This issue was raised in previous review articles, for example, Houldcroft and Breuer, Nat. Rev Microbiology 2017. However, it remains unsolved.
I am grateful for the valuable recommendation of the Reviewer.
I agree that one of the major barriers of sample preparation for metagenomics is the high abundance of host nucleic acids in the samples, which when extracted and sequenced lead to significant diminishment of the viral reads that are generally with very low presentation. For this reason, viral concentration techniques and removal of host genetic material are crucial for the success of the metagenomic analysis. Due to this, I added a new paragraph in the section “Challenges of metavirome analysis in regards of blood donation”, citing the suggested article Houldcroft and Breuer, 2017, Nat. Rev Microbiology. The following information was added:
Lines 193-206, Page 5: “... Viral metagenomics is related to multiple laboratorial challenges especially sample preparation but one of the most important includes the high presence of contaminat-ing host nucleic acids in the extracted samples that can significantly impact the sensi-tivity of viral identification. For such reason host-depletion and viral concentration techniques (filtration, ultracentrifugation, addition of nucleic acid carriers) must be mandatory applied during laboratory procedures with samples destined for meta-genomics [43]. Although such depletion/concentration procedures have beneficial im-pact on host nucleic acids removal from clinical samples they can also lead to dimin-ishment of the viral genomes of interest reflecting the preparation of sequencing libraries [43]. The presence of host nucleic acids represents significant challenge in the metagenomics, therefore different laboratory techniques that are directed towards di-minishment of host genetic material must be comprehensively compared and these ones providing the best ratio between host nucleic acid decontamination: viral genome acid preservation might be further established as laboratory standards for sample management. …”
Capture sequencing could increase the sensitivity but it works only for known viruses. Increase of sequencing depth could be a partial solution but it makes the contamination worse. There are reports of other methods such as enzyme treatment and ultracentrifuge. However, we don’t know if these methods really work because of the lack of comparison. While this review manuscript is specially for blood donation, the dominance of host nucleic acid in serum/plasma is a root cause of NGS detection sensitivity and contamination. Therefore, I suggest Dr. Slavov to mention this important point in appropriate sections, for example, section 4 (Challenges …).
I am grateful to the reviewer for the valuable suggested that helped to improve the paragraph dealing with the host contaminants. The following explanations were added:
Lines 201-206, Page 5: “...The presence of host nucleic acids represents significant challenge in the meta-genomics, therefore different laboratory techniques that are directed towards diminishment of host genetic material must be comprehensively compared and these ones providing the best ratio between host nucleic acid decontamination: viral genome acid preservation might be further established as laboratory standards for sample management.
Minor: Proof reading may be necessary. For example, line 267: “….fom the low viral loads of ……..” should be “…..from the load loads of …….” Because contaminants are not necessarily viruses.
I am grateful for the valuable recommendation of the reviewer. Proof-reading was performed. The suggested correction was also introduced in the modified version of this manuscript.

Reviewer 2 Report
This is a review article on the emerging field of viral metagenomics applied to blood donation which aims to summarise the use and challenges of viral diseases that can threaten blood safety.
The review does acknowledge that any whether an infectious agent is a threat to blood safety is complex and depends on the ability to cause clinical infection in recipients and stability and transmissibility.
However, the review struggles in the language used and falls short evaluating any future role whether of systematic metagenomic testing for surveillance is warranted in blood donors, as it concludes it is a useful tool for continuous surveillance of future blood threats.
It uses HIV as the example as to why there is a dramatic threat and vulnerability but also notes that viruses can be identified within hours, so the context is not justified.
Three key factors are generally considered for a blood safety risk; the evidence for transfusion-transmission, the prevalence of infectious viraemia and the severity of infection. Viral metagenomics can not determine any of these factors and may result in more questions than answers at considerable cost.
It used several arboviruses as an example that are TT- Zika, JEV and RRV but Zika virus testing without considering infectivity and clinical impact (all cases of TT were asymptomatic) demonstrates the issues of sensitive screening without context.
The study in Brazil on metagenomics testing concluded ‘the crucial importance of the donor providing a timely PDIR for the prevention of transfusion transmission of viral infections which are not routinely tested in the blood banks worldwide’. This does not conclude that testing should occur but rather that post donation illness reporting is effective at preventing transfusion-transmission. Notably even during large dengue outbreaks blood donor screening has not been introduced.
Blood donor screening must consider the clinical impact of a transfusion-transmitted cases dependent on the underlying risk in asymptomatic blood donors but this requires infectious virions above the minimal infectious dose, that survive blood storage techniques and result in infection. However, risk based decision making should now be foremost at any decision on blood safety including surveillance which includes health economics data and operational assessments before introduction. The review does not capture all these issues and based on the evidence that detection of viral genetic material is insufficient to evaluate blood safety threats costs and operational impacts should be considered. It is a long bow to draw that viral metagenomics continued surveillance would avoid an event like HIV, we have seen with emerging threats to blood safety that impacts in modern times have not been comparable and before recommending such as conclusion the costs and issues should be further expanded on.
To be suitable for publication it is my conclusion this would need a significant rewrite and it is not suitable in its current form given the conclusion that it is a useful tool for blood safety has not been fully evaluated before the conclusion was made, especially in the context of systematic surveillance rather than exploratory research based.
Author Response
RESPONSE TO THE REVIEWER COMMENTS/CRITICISMS
REVIEWER #2
This is a review article on the emerging field of viral metagenomics applied to blood donation which aims to summarise the use and challenges of viral diseases that can threaten blood safety.
The review does acknowledge that any whether an infectious agent is a threat to blood safety is complex and depends on the ability to cause clinical infection in recipients and stability and transmissibility.
However, the review struggles in the language used and falls short evaluating any future role whether of systematic metagenomic testing for surveillance is warranted in blood donors, as it concludes it is a useful tool for continuous surveillance of future blood threats.
I am grateful for the valuable comment of the reviewer.
I agree that in the conclusion section, the future directions of the metagenomics in the filed of blood transfusion are not well presented. Viral metagenomics is the most suitable tool for viral discovery and especially for identification of sequences of unknown or poorly characterized viral agents. As such, it can also be applied in hemotherapy and especially for discovery of emerging threats that might impact blood donation. This is especially valid for the tropical regions where the emergence of new viruses (or presence of frequent arboviral outbreaks) show higher incidence. At its current stage, the introduction of next generation sequencing as a routine surveillance tool for continuous genomic surveillance of viruses that might impact transfusion safety is not warranted mainly due to challenges related to sample preparation, bioinformatic analysis and cost of the reactions. The following information was added to the section “Final remarks and conclusions”:
Lines 308-320, Page 8: “... Currently, at this stage viral metagenomics is best applied for the identification of un-known or poorly characterized agents. Nevertheless, implying transfu-sion-transmission potential of such viruses is difficult due to challenges related to sample preparation (mainly viral concentration, abundance of host nucleic acids) and the bioinformatic profile of the blood virome (presence of commensal viruses and con-taminants among others). Moreover, due to the cost, laborious preparation and analy-sis of such reactions, viral metagenomics is restricted to specific situations where a transfusion-transmission of emerging agent is highly suspected rather than routine use in the hemotherapy practice. However, in tropical areas or regions with high potential of explosive viral outbreaks and/or virus emergence that can impact blood donor pop-ulation, viral metagenomics can be applied in terms of identification of potential vi-ruses that might impact blood transfusion process especially considering the experi-ence with DENV or ZIKV during outbreaks. …”
It uses HIV as the example as to why there is a dramatic threat and vulnerability but also notes that viruses can be identified within hours, so the context is not justified.
I am grateful for the valuable comment of the reviewer.
HIV was given as an example underlining the vulnerability of the blood transfusion systems to emerging viruses. The emergence of this retrovirus in the 80s changed the policies of blood donation and implementation routine testing for infectious diseases (concepts like immunologic windows, specificity and sensitivity were highly discussed) in the Blood Banks worldwide. It also demanded considerable time and efforts to identify this virus and especially its blood transfusion potential. HIV-1 was characterized for an approximate period of three years (Schnid S, 2018). Anyway, by the application of viral metagenomics, it is possible to genetically characterize a given virus within hours due to the advent of the NGS techniques and the bioinformatic analysis. The following modification was performed in the manuscript text:
Lines 300-306, Page 7: “... The HIV emergence in the early 80s is a classic example underlining the vulnerability of the transfusion system to emerging viral threats [55]. HIV was characterized within three years demanding international cooperation and significant efforts. Nowadays, with the advent of NGS and bioinformatics the genomic characterization of a given viral agent in clinical materials takes hours. Therefore, in the near future, NGS can be regarded as a method of choice for molecular diagnosis, vaccine development and new therapeutic strategies [56] …”
Three key factors are generally considered for a blood safety risk; the evidence for transfusion-transmission, the prevalence of infectious viraemia and the severity of infection. Viral metagenomics can not determine any of these factors and may result in more questions than answers at considerable cost.
I am grateful for the valuable comment of the reviewer.
The confirmation of the transfusion potential of a given viral genome identified by metagenomics is a complex issue and depends on many issues like prevalence of infectious viremia and clinical impact of this agent in the transfused recipient. This was addressed in this minireview under the following lines:
Lines 276-283, Page 7: “... Even more challenging is establishing transfusion transmission properties of the iden-tified by metagenomics viral agents in blood samples (Figure 1). Even if we assume that a genome of unknown virus is assembled and taxonomically classified, there is a long way to walk until this agent enters in the list of potential emerging transfusion threats. For example, viral nucleic acids of respiratory viruses like influenza A [42] and SARS-CoV-2 [52] have been detected in plasma samples from blood donors but their transfusion transmission is probably ineffective [53] and as a result their importance for the transfusion medicine is limited. ….”
Lines 284-297, Page 7: “... In order for a virus genome identified by metagenomics to be considered transfu-sion-transmitted it must obey with several issues. One of the most important criteria is that the virus must cause a recognizable clinical impact in recipients. For example, the anelloviruses and HPgV-1 that are a normal component of the human virome are frequently transmitted by blood transfusion. However, they cause no apparent clinical disease in blood recipients. Another important characteristic is that the virus must remain viable in collected blood and must be infectious when introduced by vein punction. Of importance is also the frequency with which the infection is transmitted to susceptible recipients (transfusion transmission rates) that depends on the duration of viremia and blood donation frequency. Potential transfusion-transmitted viruses must also be evaluated in regards of viremia kinetics, immune responses and window periods. [54]. It is obvious that in its current form the viral metagenomics cannot respond to none of the above mentioned criteria, however, a rapid metagenomic identification can suggest viral agents that can be further recognized as a possible transfusion-transmission threats. …”
I am completely aware that viral metagenomics cannot respond the criteria for one indetified virus to be transfusion-transmitted. However, it can be used as a valuable tool for virus genomic identification (screening) in the light of blood transfusion, that can be further suggested as a transfusion-transmissible agent. In that respect, NGS and viral metagenomics could give a rapid response of the type of infectious agent, is genomic organization and taxonomy but of course further characterization of the transfusion potential of that virus is mandatory. I added one additional part in the part “Challenges of the metavirome analysis in regards of blood donation”.
Lines 294-297, Page 7: “... It is obvious that in its current form the viral metagenomics cannot respond to none of the above mentioned criteria, however, a rapid metagenomic identification can sug-gest viral agents that can be further recognized as a possible transfusion-transmission threats. …”
It used several arboviruses as an example that are TT- Zika, JEV and RRV but Zika virus testing without considering infectivity and clinical impact (all cases of TT were asymptomatic) demonstrates the issues of sensitive screening without context.
I am grateful to the valuable comment of the reviewer.
I agree with the reviewer that the possible transfusion-transmitted case of ZIKV was related to no clinical consequences (Motta IJ et al., 2016). However, considering Japanese encephalitis virus (JEV), one of the patients who received blood transfusion developed severe encephalitis and lethal outcome (Cheng VCC et al., 2018). Clinical symptoms were also observed in the blood recipient of the blood unit positive for the Ross river virus (Hoad VC et al., 2015). I added the following explanation in the manuscript:
Lines 95-99, Page 3: “... Transfusion-transmission of other arboviruses related to clinical consequences were reported for the Japanese encephalitis virus (JEV), where the blood recipient developed severe encephalitis and death [17] and for Ross river virus, where the transfusion transmission was related to a symptomatic infection [18]. ZIKV transfu-sion-transmission has also been reported but with no apparent clinical effects [19]....”
The study in Brazil on metagenomics testing concluded ‘the crucial importance of the donor providing a timely PDIR for the prevention of transfusion transmission of viral infections which are not routinely tested in the blood banks worldwide’. This does not conclude that testing should occur but rather that post donation illness reporting is effective at preventing transfusion-transmission. Notably even during large dengue outbreaks blood donor screening has not been introduced.
I am grateful for the valuable comment of the reviewer.
The introduction of a paragraph that describes metagenomic analysis on blood donors reporting post donation illness, was to outline a situation, where there is evidence that pathogenic viruses might be identified in blood donor by metagenomics. This section is introduced long after the presentation of the normal blood virome observed in healthy blood donors, where predominantly commensal viruses with limited impact to blood transfusion are considered. With other words, the application of viral metagenomics to symptomatic donors, gives us suitable information for the pathogenic viruses that might be transmitted by blood transfusion and as a consequence. I added the following explanation in this part of the manuscript:
Lines 170-173, Page 5: “... In such cases, the application of viral metagenomics on these samples can give valua-ble information in respect of the PDI causative agents and is a suitable route to identify viruses that can impact transfusion safety. …”
Lines 174-182, Page 5: This paragraph was generally reformulated:
“...Brazilian metagenomic studies on plasma samples from blood donors reporting PDI symptoms including fever, retro-orbintal pain, exanthema, headache, myalgia, diar-rhoea, respiratory symptoms and jaundice was performed. Viruses like DENV-2, par-vovirus B19 (B19V) and even influenza virus (H3N2) were identified by viral meta-genomics in several samples [41,42]. In this respect, application of metagenomics in blood donors who consequently presented symptomatic diseases reveals the potential of metagenomics to unveil viruses with potential impact on transfusion safety and shows the importance of PDI for prevention of transfusion transmission of infectious agents that are not routinely tested by the Blood Transfusion Services. …”
This following text was removed from the manuscript:
“... Both of the identified viruses are transmitted by blood transfusion and cause clinical impact in the affected populations. Application of metagenomics in such cases provides valuable information of transfusion threats that are not routinely screened by the blood transfusion services. Beyond typical blood-borne viruses, , in plasma of blood donors who relate postdonation information have been identified also respiratory viruses, like influenza virus (H3N2) [41]. This is a rare finding, and the postdonation information was characterized by typical respiratory symptoms. While it is unclear if respiratory viruses with short and low viral load viremia can be transmitted by blood transfusion, the above mentioned studies show the enormous potential of viral metagenomics to unveil viral transfusion threats in blood donors. This information is crucial for donor counselling to provide postdonation information as to prevent the transfusion transmission of emerging viral threats. It can also be suitable for blood donation institutions and their policies regarding the implementation of rigorous system for tracking and dealing with postdonation illness events reported by the blood donors. …”
Lines 183-187, Page 5: This text was completely modified “... An important observation is that the above cited studies were performed in tropical regions, where intensively circulate many arboviral and emerging viruses. Ideally, a global multicentric metagenomic study including blood donors who report PDI from different climatic areas, is the best option to unveil a broad variety of viruses capable of causing symptoms in blood donors and with probable transfusion transmission. …”
Blood donor screening must consider the clinical impact of a transfusion-transmitted cases dependent on the underlying risk in asymptomatic blood donors but this requires infectious virions above the minimal infectious dose, that survive blood storage techniques and result in infection. However, risk based decision making should now be foremost at any decision on blood safety including surveillance which includes health economics data and operational assessments before introduction. The review does not capture all these issues and based on the evidence that detection of viral genetic material is insufficient to evaluate blood safety threats costs and operational impacts should be considered. It is a long bow to draw that viral metagenomics continued surveillance would avoid an event like HIV, we have seen with emerging threats to blood safety that impacts in modern times have not been comparable and before recommending such as conclusion the costs and issues should be further expanded on.
I am extremely grateful for the valuable comment of the reviewer.
Taking the consideration of the reviewer, the conclusion was totally reformulated with presentation of the advantages and disadvantages of the metagenomics in especially blood transfusion aspect and accentuation on the fact that a simple identification of a virus genome could not be considered transfusion threat. Thus the conclusion of the manuscript was totally reformulated and I included the following conclusion part:
Lines 321-342, Page 8: New conclusion was added:
“... It is of utmost importance to provide the safest blood products that are free of infec-tious agents. The revolutionary development of NGS and bioinformatic analysis in the face of metagenomics has also been applied to blood donor samples thus able to reveal their genetic content including all viruses present. This have raised questions regard-ing the presence of emerging viral agents that can impact transfusion safety. In that line, metagenomics can be successfully used as a rapid and efficient approach to iden-tify a viral genome especially when the virus is poorly described or completely un-known. Although discovery of viral nucleic acids can shape our opinion for future transfusion threats it is not sufficient to imply that one genetic sequence could have transfusion transmission properties. Several issues, the most important of which in-clude the clinical impact of the virus in the recipient, viral survival in the stored blood, viral load capable of causing infection and efficiency of virus transmissibility by blood must be thoroughly investigated. This argues against the routine application of viral metagenomics as a tool for viral infectious surveillance in the Blood Banks worldwide. In that line, risk-based decision making for blood safety should be the most important tool combing health economics and operational assessments before introduction of any NGS procedure. Although viral metagenomics could give us insights on the possible emerging transfusion threats, it is insufficient only by genomic characterization to re-veal viruses that can impact transfusion medicine. The balance between economic costs and the most important health priorities of given country will further shape the possible application of viral metagenomics as a routine tool for identification of emerging viruses in the field of blood transfusion.…”
To be suitable for publication it is my conclusion this would need a significant rewrite and it is not suitable in its current form given the conclusion that it is a useful tool for blood safety has not been fully evaluated before the conclusion was made, especially in the context of systematic surveillance rather than exploratory research based.
I am grateful for all of the points that were raised by the reviewer. All of them were taken when performing the revision of the text. The conclusion was completely modified as discussed in the above response.

Reviewer 3 Report
This is a generally well developed review of the broad area of application of viral metagenomics to blood donations. There is redundant throughout the manuscript that would allow for reduction in length by 10-20%. The English language also needs careful attention.
Specific Comments"
Introduction:
No need to (wrongly) use to the word "virosphere", and "...identification of even the lower viral loads...." is a stretch since NGS detection still few logs lower than PCR depending on type clinical sample and virus and few are optimizing for blood. I am only aware of extensive optimization of sensitivity for CSF.
Line 72. The fact that NGS can be used on blood does not raise question about the safety of blood. Just provides evidence for nucleic acids in blood and potentially another means to test for infectious agents that are determined to be sufficiently prevalent in healthy donors, transmissible by transfusions and to consequently cause clinical disease in recipients.
Line 118. There are more than one human pegivirus.
Line 167. Other publications of donor populations report much higher pegivirus prevalence.
Line 204. Reference 41 is an exception; most other reports are negative.
Good points about common contamination unless extreme precautinary measures are taken. This problem is nicely reviewed in very recent NEJM publication: Simner and Salzberg. The Human “Contaminome” and Understanding Infectious Disease. N Engl J Med 387;10:943-946 September 8, 2022
Author Response
RESPONSE TO THE REVIEWER COMMENTS/CRITICISMS
REVIEWER #3
This is a generally well developed review of the broad area of application of viral metagenomics to blood donations. There is redundant throughout the manuscript that would allow for reduction in length by 10-20%. The English language also needs careful attention.
I am very grateful for the positive feedback of the reviewer regarding the presented manuscript. I have tried to exclude the redundant writing and many parts of the manuscript were corrected. The corrected version of the manuscript, I believe does not present much redundancy. Once again thank you very much!
Specific Comments"
Introduction:
No need to (wrongly) use to the word "virosphere", and "...identification of even the lower viral loads...." is a stretch since NGS detection still few logs lower than PCR depending on type clinical sample and virus and few are optimizing for blood. I am only aware of extensive optimization of sensitivity for CSF.
I am grateful for the valuable comment of Reviewer #3.
The explanation in parentheses “(also called virosphere) was removed from the revised version of the manuscript.
I agree with the reviewer that NGS detection presents still with lower sensitivity for the detection of some viral agents compared to the traditional molecular techniques (nested-PCR for example). This phrase was presented in the manuscript due to the fact that the current production platform sequencers (NovaSeq for example) have greatly improved their sequencing depth and the possibility to detect low viral loads. I removed from the text “even the lowest viral loads”. The phrase reads now like this:
Lines 47-48, Page 2: “... . The high depth of sequencing permits the identification of viruses with low representation in the tested samples, especially the emerging ones (Figure 1)....”
Line 72. The fact that NGS can be used on blood does not raise question about the safety of blood. Just provides evidence for nucleic acids in blood and potentially another means to test for infectious agents that are determined to be sufficiently prevalent in healthy donors, transmissible by transfusions and to consequently cause clinical disease in recipients.
I am very grateful for the valuable comment of the reviewer.
I agree that the application of NGS and metagenomics on blood samples provides evidence for the genetic content of such samples (viral abundance) and can determine possible infectious agents that can be hypothetically transmitted by blood transfusion. I performed modification in this part of the text, which is presented now like this:
Lines 75-79, Page 3: “...The application of NGS and metagenomic analysis on blood samples can provide in-sights for the viral nucleic acid abundance in such specimens. In this respect, the high potential of metagenomics for virus discovery can represent a suitable means for iden-tification of poorly described or emerging viruses in blood donors or hemoderivatives that can be regarded as possible transfusion threats. …”
Line 118. There are more than one human pegivirus.
I am grateful for the valuable comment of the reviewer.
Human pegivirus-2 (HPgV-2) is a new classification name of the Human Hepegivrus-1. An excellent review regarding this viral agent is written by Chen S et al., 2022 (Viruses). I presented this type of virus in the manuscript, however, I am not very sure for its participation as a normal component of the human blood virome as it has been very rarely detected in healthy subjects. The following phrases were added to the revised version of the manuscript.
Lines 116-117, Pages 3,4: “... Human Pegivirus-2 (HPgV-2) and probably many other not identified viruses which are...”
Lines 156-159, Page 4: “... Another non-pathogenic human pegivirus, HPgV-2 with similar to genomic organiza-tion to HPgV-1 was also described in 2015. Although its prevalence is still unclear, it seems that it is infrequent in the healthy population [24]. Further metagenomic studies will reveal if HPgV-2 is also a normal component of the blood virome.…”
Line 167. Other publications of donor populations report much higher pegivirus prevalence.
I am very grateful for the valuable comment of the reviewer.
The prevalence of HPgV-1 is highly variable depending on the location and population tested. The cited article (Yang et al., 2020), is a recent systematic review and metanalysis that evaluates the global prevalence of HPgV-1 among blood donors. Based on the random effect metanalysis of 35,468 blood donors the established prevalence of HPgV-1 was 3.1% with pooled prevalences per location like follows 1·7% (95% CI, 1·1-2·6) in North America, 9·1% (95% CI, 6·4-12·7) in South America, 2·3% (95% CI, 2%, 2·8) in Europe and 2·4% (95% CI, 1·4-4) in Asia. The following modification was performed in the manuscript:
Line 153, Page 4: “… globally reaching 3.1% …”
Line 204. Reference 41 is an exception; most other reports are negative.
I agree that finding influenza virus A RNA in blood donor reporting post donation illness is an exception, but it shows the potential of metagenomics to unveil unsuspected viruses in human plasma. In fact, all of the studies that seek to evaluate the prevalence of influenza A RNA in blood donors, even during large influenza A epidemics are negative (Stramer SL et al., 2012; Sobata R et al., 2011). Even though influenza A was found by metagenomics in human plasma, it transfusion transmission potential is probably insignificant as it is respiratory virus and blood is not an usual route of virus transmission.
Good points about common contamination unless extreme precautinary measures are taken. This problem is nicely reviewed in very recent NEJM publication: Simner and Salzberg. The Human “Contaminome” and Understanding Infectious Disease. N Engl J Med 387;10:943-946 September 8, 2022
I am grateful for the excellent reference suggested by the reviewer. It was accordingly cited in the revised version of the manuscript and I added the following paragraph considering the “computational contaminants”:
Lines 235-242, Page 6: “… The so called “computational contaminants” can also affect the metagenomic analysis. The presence of small fragments of human DNA contaminants in assembled microbial sequences has been reported. This finding makes challengeable the application of metagenomics for identification of infectious agents in human samples, once the bio-informatic analysis compares the obtained from human specimens sequence reads to known viral, bacterial, parasitic or fungal sequences that might be implied as cause of infection. The exclusion of contaminant reads and identification the cause of infection is such situations is fundamental to avoid misdiagnosis [47]. …”

Round 2
Reviewer 2 Report
I thank the author for thoroughly addressing my concerns and it is now in a state to be published with some minor additional editing that I assume the editing process will also help with.
There are some inappropriate capitals and wording such as "long way to walk' and 'obey with several issues' that would need editing.
I suggest replacing some 'can' with 'could' e.g viral metagenomics could be applied in terms of identification given the uncertainty.
The introduction of risk based decision making in the conclusion needs a reference. Either the ABO document or peer review publications on the RBDM framework in Vox Sanguinis or Transfusion would be appropriate.
Author Response
REVIEWER #2
I thank the author for thoroughly addressing my concerns and it is now in a state to be published with some minor additional editing that I assume the editing process will also help with.
I am very grateful for the raised concerns that significantly improved the presentation of the manuscript and its content. I also believe that the editing process will further help to improve the style of this paper. Thank you very much!
There are some inappropriate capitals and wording such as "long way to walk' and 'obey with several issues' that would need editing.
I am very grateful for the valuable comments.
I performed the following modifications:
Lines 277-280, Page 7: “long way to walk” was excluded from the text. The modified phrase sounds now the following: “… Even if we assume that a genome of unknown virus is assembled and taxonomically classified, multiple studies must be performed until this agent could be classified as a potential emerging transfusion threat.. …”
Lines 284-285, Page 7: “obey with” was replaced with “respond to”. The phrase now sounds like this “… In order for a virus genome identified by metagenomics to be considered transfusion-transmitted it must respond to several issues. …”
Some typographical errors and capital letters were also corrected.
I suggest replacing some 'can' with 'could' e.g viral metagenomics could be applied in terms of identification given the uncertainty.
We are grateful for the valuable suggestion of reviewer#2
In the text, can was used 41 times. I changed can to could in 20 situations in order to express uncertainty.
The introduction of risk-based decision making in the conclusion needs a reference. Either the ABO document or peer review publications on the RBDM framework in Vox Sanguinis or Transfusion would be appropriate.
I am grateful for the valuable suggestion of reviewer#2
The article by Janssen MP et al., 2020 published in Transfusion was cited in the context suggested by reviewer #2.
“Janssen MP, Nuebling CM, Lery FX, Maryuningsih YS, Epstein JS. A WHO tool for risk-based decision making on blood safety interventions. Transfusion. 2021 Feb;61(2):503-515. doi: 10.1111/trf.16231. Epub 2020 Dec 25. PMID: 33368381; PMCID: PMC7898802.” (reference number 57).
Reviewer 3 Report
The revision addresses the concerns and recommendations that I provided and those of the other reviewers.
Author Response
REVIEWER #3
The revision addresses the concerns and recommendations that I provided and those of the other reviewers.
I am very grateful for the positive opinion of reviewer #3 on the revised manuscript. The comments raised improved significantly the presentation of the information and the manuscript in general. Thank you very much!